# Digital Identification of the Human Condition as a Prerequisite for the Effectiveness of the Organizational Automation (Biocybernetic) Systems Operation

**DOI:** 10.3390/s22103649

**Published:** 2022-05-11

**Authors:** Vladimir L. Kodkin, Ekaterina V. Artem’eva

**Affiliations:** 1Power Engineering Faculty, South Ural State University, 454080 Chelyabinsk, Russia; 2Department of Chemistry, Institute of Natural Sciences, South Ural State University, 454080 Chelyabinsk, Russia; artemevaev@susu.ru

**Keywords:** human-machine interaction, biocybernetic, structural analysis, ECG, ECG isoline, the physiological state

## Abstract

The article deals with the problems of improving modern human-machine interaction systems. Such systems are called biocybernetic systems. It is shown that a significant increase in their efficiency can be achieved by stabilising their work according to the automation control theory. An analysis of the structural schemes of the systems showed that one of the most significantly influencing factors in these systems is a poor “digitization” of the human condition. “Digitization” here is the identification of a person as a participant in the interaction with a cybernetic or cyber-physical system. The main problem of a biocybernetic system construction is the non-stationarity of such human characteristics as time of the reaction to external disturbances, physical or nervous fatigue, the ability to perform the required amount of work, etc. At the same time, as a rule, there is no objective assessment of this non-stationarity. Under these conditions, ensuring the controllability and efficiency of biocybernetic systems is a very difficult task. It is proposed to solve this problem with the help of electrocardiogram signals: the most accessible and accurate information about a human’s current state. Herein, several examples of such solutions and the results of theoretical studies and experiments are discussed.

## 1. Introduction

At present, the interaction of a person and various technical means (“human-machine” interaction) is reaching a new level. For several decades, “human-machine” interaction has been understood as the joint functioning of human “natural” intelligence and “artificial” intelligence of computational and automatic systems [1,2]. In this interaction, as a rule, the main attention was paid to increasing the intelligence of machines and the ability to independently solve an increasing range of tasks, while maintaining the advantages of artificial intelligence: reliability, accuracy, and speed of information processing.

Most often, this interaction had a pronounced “hierarchical” structure. The person set tasks for the technical unit and controlled their implementation. Such intervention in the course of the task execution had to be very limited. One of the automatic system tasks is to work as long as possible without the participation of an individual person and their intelligence. Significant progress has been achieved in this direction. Moreover, such a situation has developed that a person with their intuitive, unstable intellect, subjected to numerous hardly identified influences, becomes a “weak” link in human-machine interaction. This problem is becoming more and more significant at a time when human-machine interactions are becoming extremely diverse.

## 2. Problem Statement

The interaction of a human and system technical means can be observed in almost all areas of activity from industrial units to simulators and medical systems, in which a person is not only an object of examination or treatment, but also a subject, managing this process. If an object is subjected to the maximum possible examination, in technical terms, “identification”, then the examiner, the person who controls the complex set of technical means, remains non-deterministic, an absolute “black box” [3]. If the technical means in terms of intelligence are still approaching a person, then a person, with their weak “digitization”, unpredictability and uncontrollability, has remained outside the development of human-machine systems.

In recent years, complex systems, called “cyber-physical” systems (CPS), have received increasing development. In such systems, cybernetic means–electromechanical elements, computers and power electronics–interact with technical units of a completely different nature. These are, for example, vehicles or technological devices. For effective design, they are “combined” with cybernetic resources into a single system. At the same time, the rapidly developing branches of personalized medicine require the combination of technical, including cybernetic, means and methods, with research on a person as a biosocial structure in the framework of solving a number of problems. It seems appropriate to formulate the concepts of “biocybernetic systems”.

## 3. Structural Analysis. General Structure

A biocybernetic system is a system in which a person and cybernetic equipment interact to achieve the goals. One of the options for the block scheme of such a system is shown in Figure 1.

In order to fulfil the task of managing the “Working mechanism”, the “Person”, represented by the central nervous system (CNS) and physiological structures, and the cyber-physical system, consisting of the automatic control system (ACS) and the actuation device controlled by this system, interact. The block diagram shows that a person forms goals through the CNS, both for “their” physiological system and for the CPS. As a result of the implementation of the tasks, the CNS makes adjustments.

The very detailed objective information about the state of the CPS as a whole and the working mechanism or the “Executive mechanism” is used. The CNS relies only on its own senses when assessing its capabilities and state. Obviously, from the automatic control theory (ACT) point of view, the structure of a person and the main elements of this structure are indefinite, non-stationary, poorly predictable and manageable compared to ACS cybernetic systems and electromechanical devices [4,5,6,7]. In this regard, the task of their joint management is quite difficult. The efficient management of such systems is to ensure that all elements of the system perform tasks with minimal expenditure of resources and time. However, relying only on information about the CPS, it is impossible to complete this task. According to the provisions of the stability theory [8], a complex non-stationary object, not “covered” by information supply, either disrupt the stability of the entire system, or significantly slow down the dynamics of all processes.

## 4. Variants of Biocybernetic Systems

An even more simplified version of the biocybernetic system is smart power simulators or exoskeletons, the schemes of which are shown in Figure 2. As a rule, in such systems, only the states of the working mechanism and, very seldom, the state of the electric drive, are controlled in order to optimize the work performed. In devices of “power” interaction, the efforts of a person and a cyber system can be opposing or joint. When a person uses a simulator, they perform physical exercise and counteract mechanical effects in order to identify their capabilities (to be tested) or to develop these capabilities (to conduct training). The tasks of such a system, from the ACT point of view, are to evaluate the accuracy of parrying loads and the quality of parrying processes.

In exoskeletons, a person, together with electromechanical devices, perform tasks, creating joint efforts. The purpose of such interaction is to complete the task with the required characteristics and optimize resources while doing the task. If we consider the block scheme of the biocybernetic system, it will become obvious that information about the state of the working mechanism is clearly not enough to ensure its controllability. External disturbances in the mechanism itself and the non-stationarity of the human characteristics greatly complicate the task.

To perform all the tasks of the “power” interaction systems, it is necessary to have information about a person as accurate and reliable as the information about the electromechanism available in modern cybernetic devices. Thus, the currents of the actuation devices, the speeds of movement of individual elements of the cyber system, the magnitude of linear and angular movements are measured in the e-transport electric drive, a simulator or an exoskeleton. Special sensors assess the capabilities of the cybernetic system and its state with high accuracy. Without similar information about a person, as a participant in the interaction, it is very difficult to create an effective, controlled biocybernetic system. However, currently, it is hardly possible to achieve such a level of accuracy in the widespread use of biocybernetic systems.

It is known from the ACT that the most necessary characteristic of feedback and control signals for the controllability of such systems is their continuity. Therefore, the information about the state of the object should be updated almost continuously. Such a signal in human-machine systems is an electrocardiogram (ECG) signal. The ECG signal is very well studied in cardiology [9,10,11,12,13]. The science of the state of the heart has studied the changes of each interval and deflection. However, it is difficult to apply this knowledge in other areas; in particular, in the biocybernetic system operation.

No matter how strange it may seem, the state of a healthy heart has been studied much less well than the bad heart state. There are many reasons for this. The main thing is that healthy people are practically not examined. The exception is athletes and people whose work is associated with physical exertion: the military, rescuers, etc. However, even in their case, monitoring of the heart is mainly aimed at fixing critical conditions. During regular medical examinations, once every 6 or 12 months, doctors are convinced that the main parameters: heart rate, R-waves, ST-segments, etc., are normal (moreover, the limits of the norms are very “wide”). This usually ends the research.

Numerous modern gadgets such as ECG T-shirts, smart watches and other similar tools have not yet given tangible results, primarily because there are no clear goals for continuous ECG recording in healthy people. The ECG is registered, sent, received, but what is the next step? If the ECG indicates a critical condition of the patient, the actions of the doctors are completely understandable: the person needs help. However, as far as we know, there are no methods to recommend to a healthy person a way to optimize their activity. Cardiology and cybernetics currently have a lot of common ground, and rarely solve common problems like those that are discussed in the article. At the same time, few observations of continuous, and non-standard ECGs lead to assumptions about their widest possibilities.

Since 2015, scientists from South Ural State University have been conducting research on the use of non-contact ECG to assess the condition of athletes [14,15,16]. Athletes were examined as “subjects”, for which the physical condition is an extremely important factor reflecting their physical activity: loads, training, and competitions. That is almost a “technological characteristic”, and a necessary condition for controlling possible boundary conditions such as overstrain, fatigue, etc.

## 5. ECG Recorders. Non-Contact Registration Device

Highly sensitive ECG recorders have been developed and produced. Their scheme was somewhat different from traditional detectors. Significant electrodes were connected only to the non-inverse inputs of the amplifiers. Through precisely matched elements, the compensating electrode was connected to the inverse inputs of analogue amplifiers that form non-standard ECG signals.

In standard recorders [17], the significant electrodes are connected to the differential inputs of the amplifiers, as shown in Figure 3a. With the new scheme, it is possible to record biopotentials at very high input resistances in the measuring circuits (active resistances in the circuits of significant electrodes can reach several mOm, very high values). In practice, this means that the signals will be recorded through a layer of natural fabric or through a not very hard contact. The main thing is that the compensating electrode (N), which is connected to the inverse inputs of the measuring amplifiers of biopotentials, should have a good contact (Figure 3b).

Direct “digitization” of the ECG signals has standard parameters: 10–12 digits in analog-to-digital conversions, and allows us to obtain an accurate identification of all the main ECG parameters. The time for generating a digital value with a duration of 1–2 ms allows us to create the necessary filters and accurately calculate all time parameters: heart rate, duration of the Q-T interval, etc. The ECG in Figure 4 is recorded through a cotton T-shirt. Large electrodes were made of an electrically conductive fabric (Figure 3c).

This design allows us to embed the electrodes in clothing and control the ECG completely without the participation of physicians, which is impossible when monitoring standard 12 leads, or when standard electrodes of very small size are installed on the patient’s body. At the same time, the information received is much superior in quality to that obtained in the well-known ECG T-shirts, the signals in which can only be used for heart rate calculations and even then are not very accurate. In addition, the options for leads found during the experiments (two on the chest, two on the back) allowed to obtain new information possibilities for a non-standard recorder as a source of continuous information about the human condition, which is very important for biocybernetic systems. The hundreds of registered cardiograms showed some peculiarities for which cardiologists and exercise physiologists could not find a simple explanation.

## 6. Results of Experimental Research. New ECG Capabilities

The main feature of the proposed means of recording the ECGs of athletes is the possibility to observe an accurate ECG directly during exercise. Figure 5 shows the ECG of an athlete working out on an exercise bike. At a specific moment of time, there was a sharp change in the R-wave, while the RR intervals almost did not change. Standard devices and techniques, as well as some fundamentally new ECG meters (“EPIC technologies”) do not respond to such changes [18,19,20].

Figure 6 and Figure 7 show the ECGs of two athletes of 24 and 40 years old: “long-distance runner” and “sprinter” registered during the training. The first one easily worked out on the exercise bike, while for the second one it was more difficult. Their ECG before the exercise almost did not differ (Figure 6).

However, after the load (Figure 7), there were a number of noticeable differences, recorded by the “traditional” physiology: the differences in heart rate were significant, the reduction in the intervals for both test people occurred due to diastole, the ECG of both athletes had ST elevations, but of different magnitude. Also, both athlete ECGs had distortions of the isoline. However, in addition, we noticed that the long-distance runner had a redistribution of negative and positive ECG impulses, while the sprinter did not. Exercise physiologists with their traditional equipment, an exercise bike with a “Schiller” monitor, did not pay attention to this “adaptation” of the cardiovascular system to loads [7,14,15,16]. New studies, made possible by the ECG recorders offered in the article, should answer the question whether distortion of the ECG isoline provides additional information about an athlete’s overstrain.

The non-contact registration device made it possible to obtain an ECG of an athlete performing strength exercises, including those associated with a change in body position (Figure 8). Figure 8 shows ECG of an athlete during body movement exercises: push-ups and slopes. ECG elements differ.

It should be noted that the ECGs were recorded when there were a significant muscle tension and dramatic changes in body position, which is impossible when registering standard ECG leads. In addition, these experiments have shown the potential to use ECG isolines as information about the state of a person during such registration.

## 7. Assessment of Human Respiration

High-precision ECG recorder connected to textile electrodes laid on bases with shape memory calculated four ECG channels:Between the right thoracic and left thoracic electrodes.Between the right thoracic electrode and the left one on the back.Between the right thoracic electrode and the right one on the back.Between the right and left electrodes on the back.

Two four-second ECG intervals are shown in Figure 9a,b.

In this recording, the ECG signal is passed through a low pass filter to eliminate isoline motion. This filter introduces some distortion into the signals, but the results are still quite interesting:The upper channel (1st) is very stable, almost does not react to movement and breathing like the 4th one.The 2nd channel contains ECG and responds to breathing,The 3rd channel does not contain ECG, but the 3rd one is highly dependent on breathing.

All figures have the same scale and time values. At 35 and 80 s the test people held their breath, and at 70–73 s, in contrast, they breathed deeply. The 1st and 4th channels display different ECG leads, the 2nd one shows a combination of ECG with chest movements, and in the 3rd one, the reaction to breathing, chest movements and, probably, internal organs, predominates.

The sensitivity of the device is sufficient to respond to both breathing and the lack of breath, the state of apnoea. All measurements were carried out with a compensating electrode directly connected to the examined person skin. A belt with textile electrodes was worn over summer clothes. Thus, registration of non-standard ECGs from the electrodes made of conductive tissues located on the back and chest of the person makes it possible to obtain information about their condition, which standard ECGs do not contain.

## 8. Biocybernetic System for Managing the Physiological State of a Person (“A Person with an Insulin Pump”)

In recent years, personalized medicine has begun to play a special role. This new medicine is based on continuous diagnostics of a human condition, consultations with specialists using IT tools and software products. In our opinion, one of the most important aspects of this technology is missing. As mentioned above, the human body is a very non-stationary structure, and the variations of certain parameters and characteristics can be very significant for different people. Even the same diseases can proceed in completely different ways and require different recommendations and treatment tactics. No doctor will be able to notice all aspects of a human condition if the examinations are carried out every 6–12 months. Therefore, the statements: “Doctors said one thing, I did another and I feel good ...” can sometimes be true and do not at all indicate the incompetence of a doctor. It is sometimes very difficult for a person to deal with themselves, and not only in terms of psychology. A special place here is occupied by chronic diseases with which a person lives for decades.

One of the world’s most common disease is diabetes. In recent years, a huge number of individual medical devices have appeared to help patients. The most complex and expensive ones are insulin pumps. The system delivering insulin to the human body can be represented by an automatic complex. Such a system contains an actuation device (an insulin pump), a complex non-stationary control object (the human body) which is influenced by disturbances, consuming incoming resources with different intensity, receiving these resources inconsistently, and an information system that receives at best, human blood sugar level, inconsistently and roughly (Figure 10) [21,22].

Units of “e” delays are especially dangerous for the work stability of the system. The lengths of delays are unique and determined by the human organism characteristics. Doctor’s “20–30 min” recommendation could be absolutely wrong [23]. The check of the glucose level control efficiency is often performed through a “diabetic cut” when the conclusion of the insulin dose correctness is made based on several measurements of blood sugar levels conducted on one day. It is clear from the scheme that the determination of time delays would be more appropriate. For that, it is necessary to carry out complex checks during few days.

1st day: blood sugar level is brought up to 10 units. Within 6–7 h, without food and insulin ingestion, the blood sugar level in the abdominal region and blood capillaries is measured every 30 min. Then the function responsible for the work of human pancreas in relation to glucose is determined.

2nd day: the sugar level is brought to 10 units and the dose of insulin (about 5 units) is injected. Within 6–7 h, without food and other insulin ingestion, the changes in blood sugar levels in the abdominal region and the capillaries of fingers are recorded. The sugar in the abdominal area is measured only for the determination of pump regulation law. This is not required for manual injection.

Based on these data it is possible to develop a methodology for the glucose blood compensation for the manual insulin injections, and the pump work plan. The task the system should solve is to ensure the variability of the glucose level in the range from 4 to 7 mmol/L. It is easy to achieve such a value for a healthy person, but very problematic for diabetics with insulin dependence. Therefore, the task for the system does not require high accuracy of the system like ACS, but requires us to ensure the minimum fluctuation of the biocybernetics structure as a whole, under conditions of limited and unreliable information security.

Ensuring controllability of the pump [11], at first glance, is not a big problem. The processes of assimilation of both sugar and insulin are rather slow. The abilities of the pump and boluses to achieve slow insulin delivery rates, and the adjustment of their values in the work process is much higher than required in terms of speed and accuracy.

The problems of the system functioning are that the human organs (where data is measured) which the resources are supplied to and processed, are completely different organs, separated (from the ACT point of view) by links with varying time delays. Processes occur in the structures with different characteristics, and these characteristics are non-stationary and non-linear. For example, automated glucose sensors measure blood sugar in the abdominal region, and glucometers measure it in the finger capillaries. Both methods have significant errors, different by nature. It takes from 30 to 50–60 min from a meal (glucose intake) to a change in blood sugar level, and from 15 to 30 min from the introduction of insulin into the subcutaneous tissues, to the interaction with glucose in the pancreas. A control system with such links stabilizes very poorly.

## 9. The Results of Studies of the Relationship between ECG and Sugar Level

Advertisements for pumps very often show blood sugar charts with an unacceptably large amplitude: from 3 to 13 mmol/L, when compensating blood sugar with insulin injections. However, a pump can lead to similar results. If we consider the structure of the “person-pump” system from the ACT point of view (Figure 10), then low rates of insulin administration are a relay link with a low output signal level. Such a link cannot stabilize a system with non-linear time delays. Moreover, it cannot even significantly affect the amplitude of these oscillations. It is known, the amplitude of oscillations in the ACS depends on the initial conditions in the dynamic system. In this case, these are the initial levels of the administered insulin and sugar doses taken. For example, if a person initially tightly controls both food intake and injected insulin, blood sugar fluctuations can be reduced.

However, firstly, with such control, there is no need for a pump. Secondly, all people have situations when it is extremely difficult to fulfil these conditions. We do not question the effectiveness of the pumps in general, we are talking about a structural analysis of this variant of the “human-machine” system. As follows from the structure, a continuous signal and the condition of a person can become a stabilizing signal. It is the ECG that can play the role of stabilizing feedback in several ways. Heart rate acceleration indicates a person’s physical activity, that is, a possible more active consumption of blood sugar, and requires a decrease in the rate of insulin supply. ST-segment shift indicates fatigue and the excessive physical activity. In addition, individual features of ECG variability are possible.

The studies also established some features of ECG changes with changes in blood sugar levels that are not “noticed” in cardiology. These are changes in the ECG isoline [23], which standard electrocardiography considers as an unfortunate misunderstanding that interferes with “correct” diagnostics. With the recommended connections of the textile electrodes, it was found that the oscillation of the isoline increases with a decrease in the sugar level. The studies were carried out both with diabetic patients and with healthy people. Figure 11 shows ECG with normal and an elevated sugar levels (differences in the pulse rate of the subject are associated with their physical test performance). The increase in the level of oscillations to significant amplitude values occurs in 30–50 s, which is significantly lower than a 20–30 min delay in the registration of blood sugar level, noted by known methods.

The use of ECG will make biocybernetic systems more efficient. The structures of biocybernetic systems take the form as in Figure 12 and Figure 13. Cross-connection by ECG and parameters calculated from ECG will significantly change the structure of the biocybernetic system. Signals can signal a state: fatigue or potential, they will be adjusted for efforts from the actuation device or insulin supply. Heart rate functionals will allow us to evaluate further actions [3,4,5,6,7]. The incorporation of continuous information about its state into a structure with a non-stationary element (a person) significantly changes the possibilities for optimizing the structure and for increase its manageability and efficiency. New options for assessing the state of a person and their potential appear.

## 10. Prospects and Hypotheses

The number of ECG cardio complexes per hour or day can be considered as a new assessment of a human state. The number of strokes is the heart rate integral. Therefore, this connection as an integral regulator will provide a high accuracy of human condition assessment, the cost of their resources. Many parameters and characteristics of a person can be associated with this variable.

The dynamic identification of a human’s state by the functional is of interest. It links the heart rate time diagram and the number of heart beats, an integral functional. The number of systems can also be compared to the work accomplished (steps), the amount of insulin injected, the volume of food, average temperature and pressure, sleep period, etc. This integral assessment can become an important characteristic of a human condition, its calculation does not require a long time or bulky expensive software and hardware.

## 11. Discussion

In “human-machine” interaction, a person performs several functions. First of all, it is the formation of tasks and control of their implementation. When monitoring and controlling interaction processes, a person tries to use as much as possible accurate information about the state of all machine structures: their “partner” in this interaction. At the same time, the state of the person remains unclear both for themselves and for the machine. It should be noted that a person experiences very significant impacts, from technical and natural to psychological ones, which most often, from the ACT point of view, cannot be identified at all. It is also very difficult to predict and determine the human response to them. These, in general, well-known provisions confirm that the problems considered in this article are extremely relevant for modern systems, which are more and more close to biocybernetic systems, the systems in which the person and the cybernetic systems jointly solve a variety of tasks.

Structural analysis shows that it is necessary to use information about the state of a person in order to provide in these systems the usual properties for cybernetic complexes: controllability, observability, stability. It is possible to ensure these qualities of the biocybernetic complex with non-stationarity and non-linearity of the “cybernetic” characteristics of a person, by increasing the information content of its functioning. “Cybernetic” characteristics of a person, including time delays, are very important for the controllability and stability of cybernetic systems. One possible method is to use the information about the state of a person, which changes at a rate commensurate with the external influences change rate. Such information includes the human ECG.

Electrocardiographic signals (ECG) are accessible, studied and rather complex information about the functional state of a person. The solutions proposed in this article and the works presented earlier for continuous non-contact ECG recording, allow us to outline the ways to solve a number of problems the specific bio-cybernetic systems face: to ensure almost continuous control over the solution of complex problems of human-machine interaction.

## 12. Conclusions

One of the most important problems complicating the creation of effective and optimal human-machine complexes is the instability and non-stationarity of the physiological functions of a person and the extremely insufficient “digitization” of the state of their central nervous system and systems of physiology.Human electrocardiography is the most accessible, accurate, informative, almost continuous information about the state of a person, which can be used in biocybernetic systems to stabilize the processes there.The experiments have shown that the ECG can be used as a source of information not only in traditional areas of diagnostics: to control heart rate, fatigue or cardiac pathological conditions, but also to analyse human conditions according to fundamentally new criteria: the number of heart beats per day, the isoline movements, sharp changes in the ECG R-waves, etc.ECG signals can play the role of stabilizing signals in non-stationary automatic systems when used in the construction of efficient and stable biocybernetic systems, the purpose of which is to perform complex technological tasks, correct a person’s condition, or perform certain physical exercises with the necessary efficiency.

## Figures and Tables

**Figure 1 sensors-22-03649-f001:**
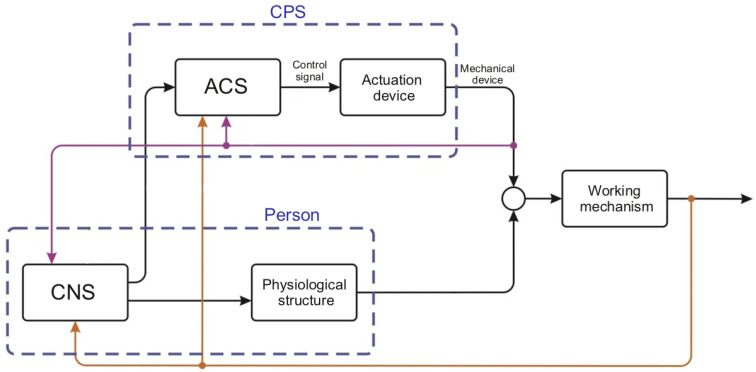
Block scheme of a biocybernetic system.

**Figure 2 sensors-22-03649-f002:**
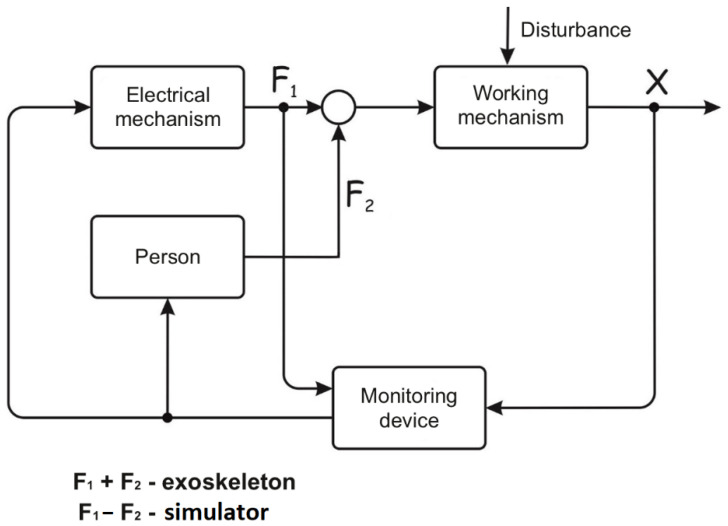
Structural diagram of the system of force interaction between a person and a “machine” with external disturbing factors.

**Figure 3 sensors-22-03649-f003:**
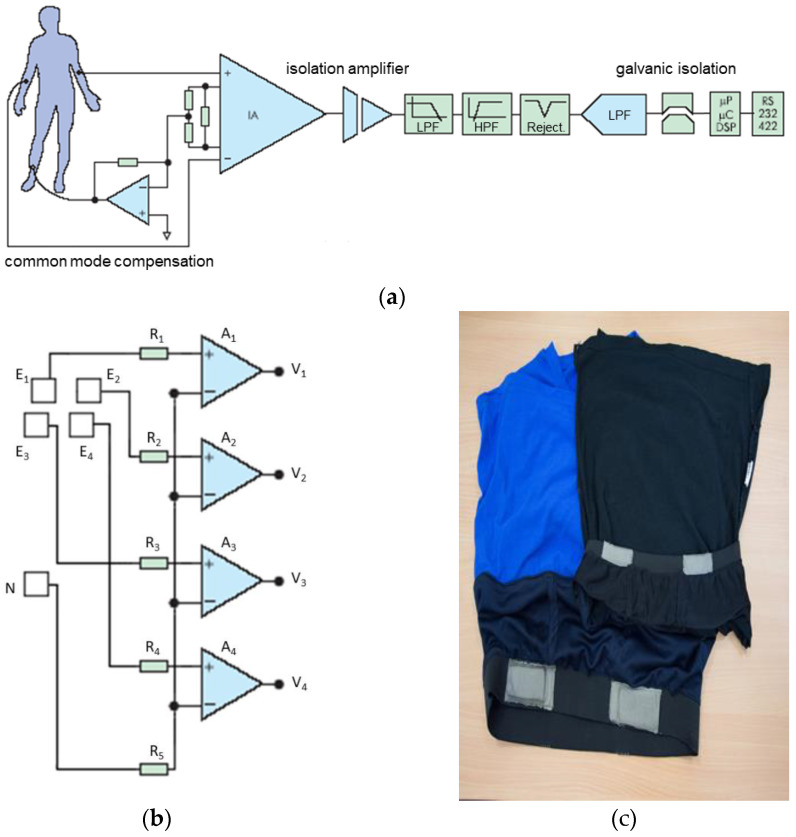
Schemes for recording a standard ECG (**a**) and non-contact ECG (**b**), clothes with electrodes were made of an electrically conductive fabric (**c**).

**Figure 4 sensors-22-03649-f004:**
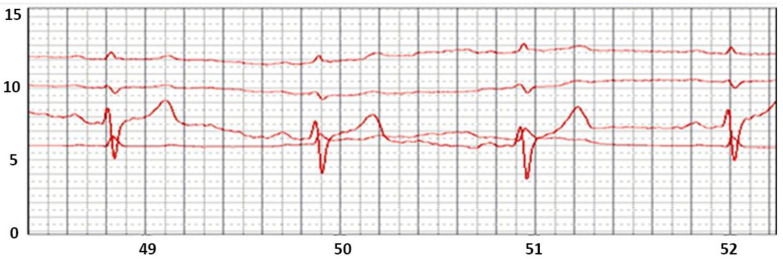
ECG from chest non-standard ECG leads recorded through clothing.

**Figure 5 sensors-22-03649-f005:**
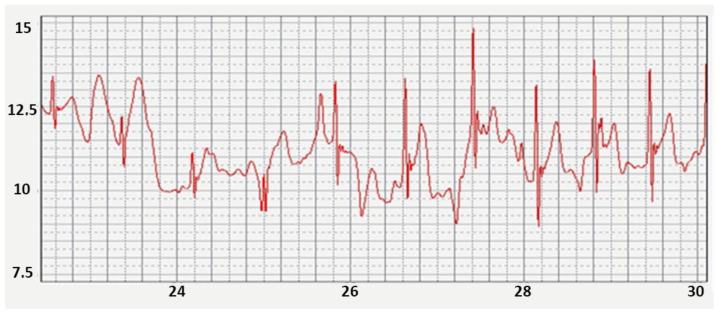
Human ECG with a sharp instantaneous change in R-wave amplitude.

**Figure 6 sensors-22-03649-f006:**
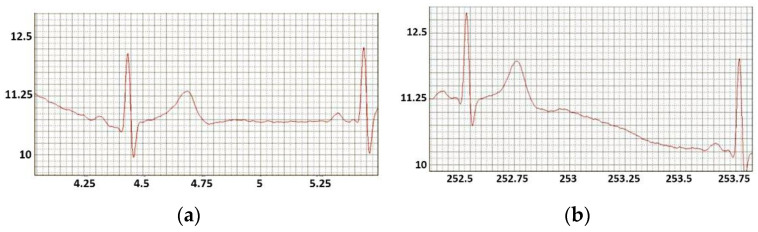
The ECG of the long-distance runner (**a**) and sprinter (**b**) before the exercise load.

**Figure 7 sensors-22-03649-f007:**
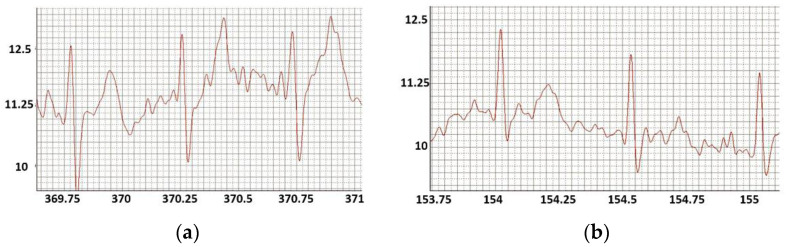
The ECG of the long-distance runner (**a**) and sprinter (**b**) during the exercise load.

**Figure 8 sensors-22-03649-f008:**
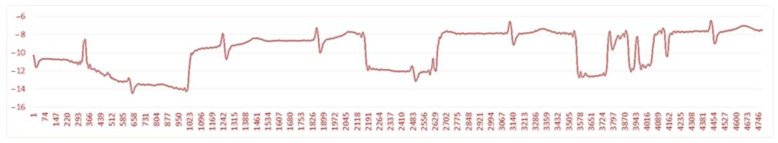
ECG of an athlete during body movement exercises.

**Figure 9 sensors-22-03649-f009:**
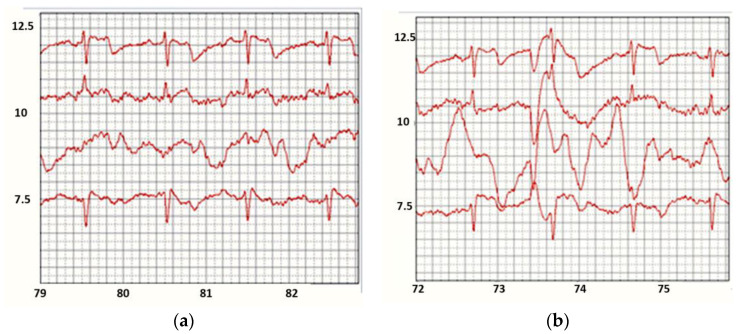
Signals from the recorder leads on the chest and back registered during normal (**a**) and deep (**b**) breathing.

**Figure 10 sensors-22-03649-f010:**
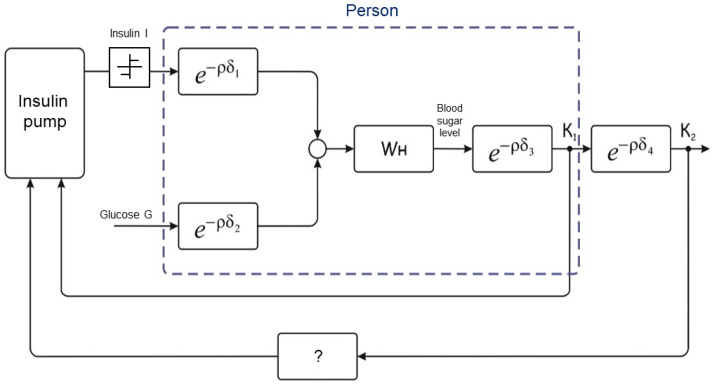
The structure of the biophysical system on the example of “a person with an insulin pump”.

**Figure 11 sensors-22-03649-f011:**
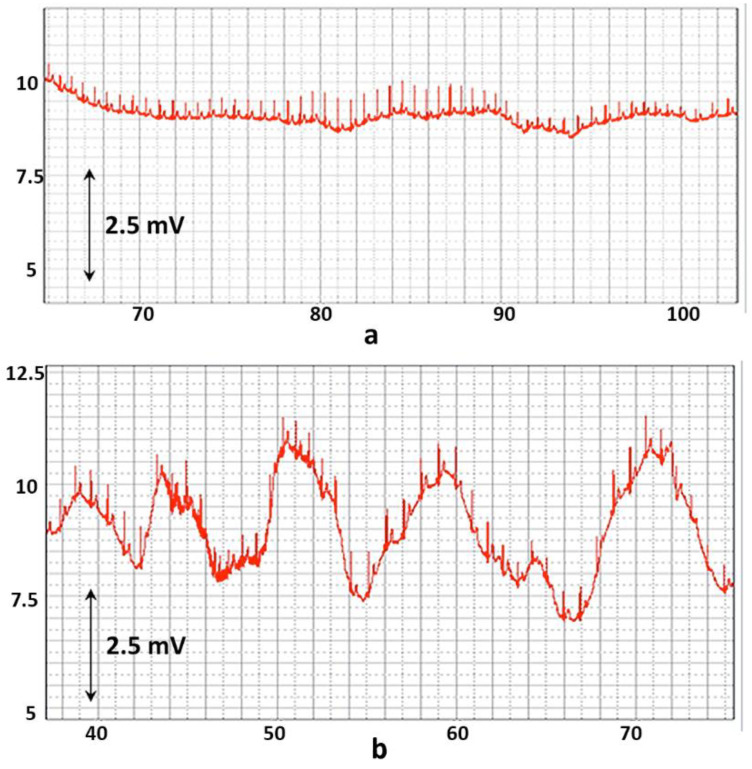
ECG of the examined patient with: (**a**) 8.8 mm, (**b**) 5.8 mm.

**Figure 12 sensors-22-03649-f012:**
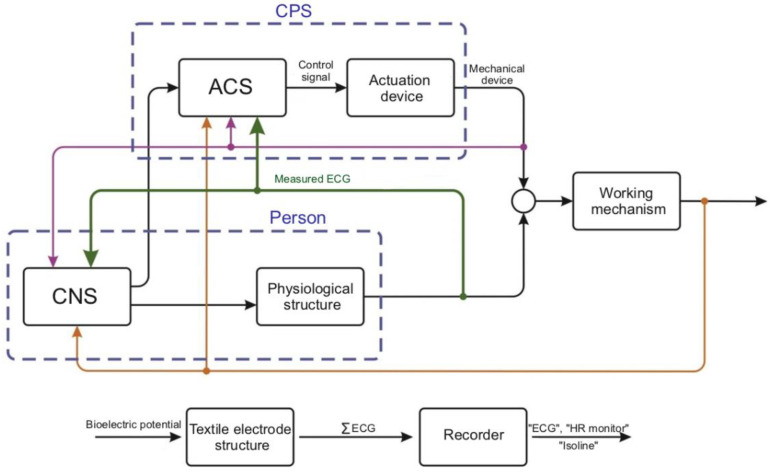
Block scheme of a biocybernetic system with an ECG link.

**Figure 13 sensors-22-03649-f013:**
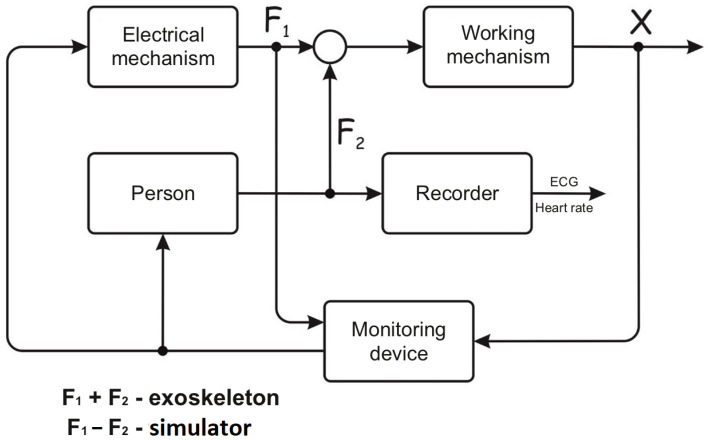
Structural diagram of the system of force interaction between a person and a “machine” with an ECG link.

## Data Availability

The raw data supporting the conclusions of this article will be made available by the corresponding author upon reasonable request.

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
