# Peer review of "Digital Identification of the Human Condition as a Prerequisite for the Effectiveness of the Organizational Automation (Biocybernetic) Systems Operation"

_sensors, 2022, doi:10.3390/s22103649_

Round 1

Reviewer 1 Report

This paper is very interesting for researchers and bio-engineers working in same research field. The authors presented the goals of this work and these goals were achieved. So, I recommended this paper for publication after revising the following minor comments:

  1. What is CPS in Figure 1?
  2. Why does the period increase in Figure 11.b?
  3. How to automatically analyze and summarize long-distance runners before or during the exercise load?

Reviewer 2 Report

This paper presents the use of electrocardiogram (ECG) signals as the identification of the human condition to increase the effectiveness of biocybernatic system. However, the paper contains several major weaknesses as listed below: 1) The title does not really show the actual content of the paper. There are many issues with the title: - "Digital identification": It is the main part of the title. However, the process of the digitization process of the human condition is not been discussed in this paper. The author suggesting to use of ECG, but they also not mention whether they are using digital or analogue ECG. - "human condition": What conditions that the authors want to see for this paper? - "prerequisite": This term is only used in the title. This actually should also be discussed in text. - "effectiveness": The effectiveness here refers to what aspect? - The title should also contains "electrocardiogram" as this is the only modality used in this work. - The title sounds general, but based on Section 9, the authors actually proposed a specific application, which is the system to monitor insulin level based on ECG. Thus, in my opinion, the title can be made towards this specific contribution. 2) The writing style need to be improved. There are many single-sentence paragraphs (i.e., paragraphs that have only one single sentence). Please expand, or combine those paragraphs. One paragraphs should have at least three sentences. 3) It is difficult to see the contribution of the paper. Based on the title, it is expected that this paper will list down several ECG characteristics (or human conditions), which are able to increase the effectiveness of biocybernetic system. However, this aspect is not been presented in this paper. Although ECG waveforms have been shown in Figures 5 to 9, there is no discussions about whether those waveforms are good waveforms or not. 4) Section 4 "Variants of biocybernetic systems." It is expected that the authors will discuss several types of the system, but in this section, only one system (simulators or exoskeletons) is discussed. 5) Figure 4 shows ECG signal from non-standard ECG leads. The authors should discussed whether this signal is good and equivalent with the signal from the standard ECG leads or not? 6) Why the authors prefer to use non-standard ECG leads than the standard ECG leads? Justifications are not given. 7) Line 114. It is not clear on what the authors mean with "has been studied much worse than the bad one."? 8) Line 121. What does "Registered, sent, received, and what is next?" means? 9) Line 151 and 152. I also not clear on what the authors want to say with this sentence. May consider to rephrase it. 10) In Section 6, how many volunteers have contributed to the data? Please specify and provide some demographic information. 11) If the authors did the data collection by their own, please provide the ethical approval information in this paper. 12) Why in Section 6, the data is taken from athletes, and not from ordinary person? What the authors aim to see from this results? 13) Figure 11 is blur. Please provide a better quality image. 14) In paragraph between line 237 and 244, the authors claim that "Doctor's "30 minutes" recommendation could be absolutely wrong." Please support this statement with citations. Also please add citations to other sentences in this paragraph too. 15) In line 15 and 372, the authors claim that the ECG provide the most accurate information about a human's current state. Please justify this statement with citation. 16) The references are lacking of journals. There are currently only 6 journals from 18 references. There is no journal from year 2021 or 2022.

Round 2

Reviewer 2 Report

I am satisfied with the corrections and justifications given by the authors.